# TASEP modelling provides a parsimonious explanation for the ability of a single uORF to derepress translation during the integrated stress response

Dmitry E Andreev[1,2†], Maxim Arnold[3†], Stephen J Kiniry[1], Gary Loughran[1], Audrey M Michel[1], Dmitrii Rachinskii[3*], Pavel V Baranov[1*]

[1]School of Biochemistry and Cell Biology, University College Cork, Cork, Ireland; [2]Belozersky Institute of Physico-Chemical Biology, Lomonosov Moscow State University, Moscow, Russia; [3]Department of Mathematical Sciences, The University of Texas at Dallas, Richardson, United States

*For correspondence:
Dmitry.Rachinskiy@utdallas.edu (DR);
p.baranov@ucc.ie (PVB)

†These authors contributed equally to this work

Competing interests: The authors declare that no competing interests exist.

**Abstract** Translation initiation is the rate-limiting step of protein synthesis that is downregulated during the Integrated Stress Response (ISR). Previously, we demonstrated that most human mRNAs that are resistant to this inhibition possess translated upstream open reading frames (uORFs), and that in some cases a single uORF is sufficient for the resistance. Here we developed a computational model of Initiation Complexes Interference with Elongating Ribosomes (ICIER) to gain insight into the mechanism. We explored the relationship between the flux of scanning ribosomes upstream and downstream of a single uORF depending on uORF features. Paradoxically, our analysis predicts that reducing ribosome flux upstream of certain uORFs increases initiation downstream. The model supports the derepression of downstream translation as a general mechanism of uORF-mediated stress resistance. It predicts that stress resistance can be achieved with long slowly decoded uORFs that do not favor translation reinitiation and that start with initiators of low leakiness.
DOI: https://doi.org/10.7554/eLife.32563.001

## Introduction

In eukaryotes, canonical translation initiation begins with the recognition of the m7G cap structure found at the 5' end of mRNAs. This is achieved by the multi-subunit eIF4F complex, which consists of a cap-binding subunit, eIF4E, a helicase eIF4A and a scaffold protein eIF4G. eIF4F then recruits the 40S loaded with eIF2*tRNA*GTP (the so-called ternary complex, TC), eIF1, eIF1A and eIF5, along with the multi-subunit scaffold eIF3 to form a preinitiation complex (PIC). Then the PIC starts to 'scan' the mRNA, unwinding mRNA secondary structures and probing mRNA for potential sites of translation initiation. After the initiation codon is recognized, the chain of events leads to large ribosome subunit joining and initiation of polypeptide synthesis. For more details see recent reviews on the mechanism of translation initiation in eukaryotes and its regulation (*Hinnebusch et al., 2016*; *Hinnebusch, 2014*; *Asano, 2014*; *Topisirovic et al., 2011*; *Jackson et al., 2010*).

Not all translation initiation events lead to the synthesis of annotated functional proteins. Many codons that are recognized as the starting points of translation occur upstream of annotated coding ORFs (acORFs) encoding functional proteins in many eukaryotic organisms (*Pueyo et al., 2016*; *Johnstone et al., 2016*; *Wethmar, 2014*; *von Arnim et al., 2014*; *Barbosa et al., 2013*; *Somers et al., 2013*; *Vilela and McCarthy, 2003*). Leader length varies greatly in mammalian mRNAs and at least 20% possess evolutionarily conserved AUG triplets upstream of acORFs

(*Churbanov et al., 2005*). The number of potential sites of translation initiation is increased further by initiation at near-cognate non-AUG codons (most frequently CUG) (*Ivanov et al., 2008 , 2011*; *Peabody, 1989*; *Tzani et al., 2016*). This is particularly prevalent for non-AUG codons located upstream of the first AUG codon, as they are the first to be encountered by the PIC (*Michel et al., 2014a*). Abundant translation initiation upstream of acORFs has been confirmed by several ribosome-profiling experiments (*Fritsch et al., 2012*; *Ingolia et al., 2011*; *Lee et al., 2012*). Ribosome profiling also revealed that the translation of these ORFs is often altered in response to changes in physiological conditions. A number of stress conditions lead to a global increase in the translation of mRNA leaders (*Andreev et al., 2015a*; *Shalgi et al., 2013*; *Gerashchenko et al., 2012*; *Ingolia et al., 2009*). Sometimes reciprocal changes in acORF translation can be observed among individual mRNAs (*Andreev et al., 2015a*). One of the stress conditions in which upstream open reading frame (uORF)-mediated translation control seems to be particularly important is in the Integrated Stress Response (ISR) (*Andreev et al., 2015b*; *Starck et al., 2016*; *Young and Wek, 2016*). The ISR induces the phosphorylation of the alpha subunit of eIF2 by one of several stress-sensing kinases. This leads to the inhibition of its recycling to eIF2*tRNA*GTP, which is carried out by recycling factor eIF2B, and to global repression of protein synthesis (*Baird and Wek, 2012*). We and others recently showed that the translation of a small number of mRNAs are resistant or upregulated during ISR, and that the most stress resistant mRNAs possess translated uORFs (*Andreev et al., 2015b*; *Sidrauski et al., 2015*).

The classical mechanism of uORF-mediated stress resistance, known as delayed reinitiation, occurs archetypically in *GCN4* mRNA in yeast (reviewed by *Hinnebusch (1993)*). Although *GCN4* regulation involves several uORFs (*Gunišová et al., 2016*; *Gunišová and Valášek, 2014*; *Dever et al., 1992*), only two are absolutely essential for stress resistance. After translation termination at a short uORF located close to the 5' end, the 40S resumes scanning albeit without the TC. The distance scanned by this ribosome subunit before the ribosome reacquires the TC depends on TC availability. Under normal conditions, most of the 40S is quickly reloaded with TC and therefore can reinitiate at a downstream inhibitory uORF. Ribosomes translating this second uORF cannot reinitiate at the acORF start. Under low eIF2 availability, a larger fraction of 40S subunits bypass the second uORF initiation codon before binding of the TC, thereby enabling acORF translation. However, examples have been found in which only a single uORF is sufficient to provide an mRNA with translational stress resistance. This has been shown to be the case for *DDIT3* (*Palam et al., 2011*; *Chen et al., 2010*), PPP1R15A (*Lee et al., 2009*), ZFAND2A (*Zach et al., 2014*), IFRD1 and PPP1R15B (*Andreev et al., 2015b*).

The start codon of the *DDIT3* uORF is in a suboptimal Kozak context and allows for leaky scanning (*Palam et al., 2011*; *Young et al., 2016*). However, the uORF encodes a specific peptide sequence that stalls ribosomes under normal conditions creating a barrier for trailing PICs, which results in strong inhibition of downstream translation (*Young et al., 2016*). It has been hypothesized that the stringency of start codon recognition is increased during particular stress conditions and that this allows for more leaky scanning (*Palam et al., 2011*; *Young et al., 2016*).

It is also possible that elongating ribosomes translating the uORF obstruct progression of scanning ribosomes downstream, and that this obstruction is relieved during stress due to reduced initiation at the uORF. Although the obstruction of scanning ribosomes may potentially explain how a single uORF can mediate stress resistance, it is unclear whether such a mechanism is plausible without additional factors involved. While most stress-resistant mRNAs possess uORFs, only very few uORF-containing mRNAs are stress resistant (*Andreev et al., 2015b*). What designates some uORFs as providers of stress resistance? To explore this, we developed a simple stochastic model of Initiation Complexes Interference with Elongating Ribosomes (ICIER) that is based on the Totally Asymmetric Simple Exclusion Process (TASEP). TASEP is a dynamic system of unidirectional particle movement through a one-dimensional lattice, in which each site can be occupied by no more than one particle and the probability of particle transition from one site to another is predefined. TASEP is widely used to model various dynamic systems, such as road traffic, and is also popular in modelling mRNA translation (*Ciandrini et al., 2010*; *Margaliot and Tuller, 2012*; *Reuveni et al., 2011*; *von der Haar, 2012*; *Zhao and Krishnan, 2014*; *Zia et al., 2011*). In ICIER, we represent scanning

and elongating ribosomes as two separate types of particles with different dynamic properties, with the possibility of transformation of one into the other at specific sites. The parameters used for the modelling were based on estimates from experimental quantitative measurements of mRNA translation in eukaryotic systems.

The application of the model demonstrates that a small subset of specific uORFs (constrained by their length and leakiness of their initiation sites) are indeed capable of upregulating translation downstream in response to the reduced TC availability that is observed under ISR. Here, we describe the computer simulations based on this model and discuss the implications of our results for understanding naturally occurring uORF-mediated stress resistance.

## Results

### The model of Initiation Complexes Interference with Elongating Ribosomes (ICIER)

The ICIER model and its parameters are illustrated in *Figure 1*. Each site of the TASEP lattice represents a codon. All sites have the same properties except two that represent the start and the stop codons of the uORF. There are two types of particles, scanning ribosomes ($\sigma$) and elongating ribosomes ($\varepsilon$). Each particle occupies 10 codons in accordance with the predominant mRNA length protected by elongating ribosomes (*Ingolia et al., 2009*; *Wolin and Walter, 1988*; *Steitz, 1969*). For

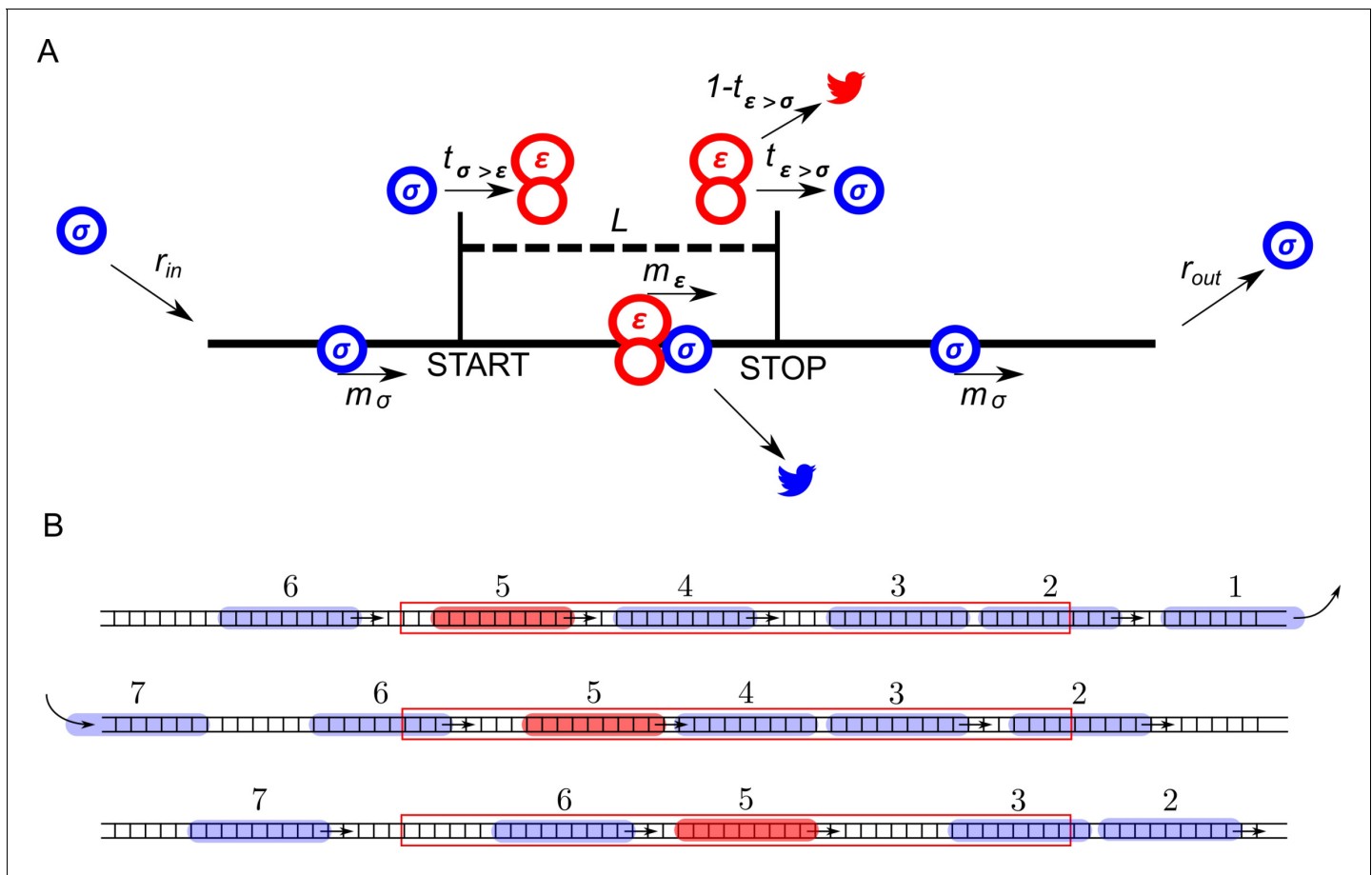

**Figure 1.** ICIER model. (A) Parameters of the model. The lattice is shown as a black line with the positions of start and stop codons separated by *L* sites (codons). Scanning ribosomes are blue and elongating are red in both panels. Ribosome dissociations from mRNAs are indicated with bird symbols. (B) An example of three subsequent states of the lattice with ribosomes shown as semitransparent oval shapes and each ribosome labeled with a unique index. Arrows indicate changes between current and subsequent steps. The red rectangle specifies the location of the uORF.
DOI: https://doi.org/10.7554/eLife.32563.002

simplicity, in our default model scanning and elongating ribosome size is the same even though scanning complexes have been shown to protect mRNA fragments of different lengths (*Archer et al., 2016*). These ribosomes can move forward by a single unoccupied site with probabilities $m_\sigma$ and $m_\varepsilon$. In addition, scanning ribosomes can transform into elongating ribosomes at the start site with a probability $t_{\sigma > \varepsilon}$, which may vary from 0 (no initiation) to 1 (non-leaky initiation). At the stop site, elongating ribosomes could transform into scanning ribosomes with a probability $t_{\varepsilon > \sigma}$ allowing for reinitiation. The remaining elongating ribosomes terminate (disappear) with the probability $1-t_{\varepsilon > \sigma}$. We hypothesized that scanning ribosomes would dissociate from mRNA when upstream elongating ribosomes collide with them. This hypothesis is based on the following considerations. When moving scanning ribosomes collide with other ribosome complexes, they may stay on the mRNA or dissociate. In the latter case, we can consider four possibilities. A scanning ribosome dissociates when it encounters a ribosome complex downstream, scanning (option #1) or elongating (option #2). Alternatively, an upstream complex (either scanning #3 or elongating #4) could cause dissociation of a scanning ribosome. The options that scanning ribosome collision cause dissociations (#1 and #3) would make formation of scanning ribosome queues impossible. However they have been observed upstream of start sites in yeast (*Archer et al., 2016*). Also, it has been demonstrated recently that an elongating ribosome pause downstream of a weak initiation site stimulates translation initiation (*Ivanov et al., 2018*). This seems possible only if scanning (and/or elongating) ribosomes could queue upstream of elongating ribosomes. This evidence makes option #2 also unlikely. Therefore, the only remaining option is that scanning ribosomes do not frequently dissociate from mRNA but they do so when the collision occurs with the elongating ribosome upstream (#4). Therefore, we used these conditions as default in our simulations. Nonetheless, we also explored how alternative scenarios (no dissociation and spontaneous dissociation of scanning ribosomes) affect the ability of uORFs to provide the stress resistance in this model (see subsection 'Miscellaneous parameters of the model: ribosome gabarits and fall off rates').

Using ICIER under different parameters, we explored how the rate of scanning ribosomes arriving at the end of the lattice $r_{out}$ depends on the rate with which scanning ribosomes are loaded at the beginning of the lattice $r_{in}$. $r_{in}$ corresponds to the rate of PIC assembly at the 5' end of mRNA, which depends on TC availability which is reduced upon eIF2 phosphorylation. $r_{out}$ corresponds to the rate of scanning ribosomes arrival to the start of the acORF. In essence, upregulation of acORF translation under stress, in terms of our model, means an increase in $r_{out}$ in response to a decrease in $r_{in}$.

## The effect of uORF length on the flux of scanning ribosomes

First, we explored how the rate of PIC loading ($r_{in}$) affects the density of scanning ribosomes downstream ($r_{out}$) of a uORF depending on its length (*L*). The results of a typical simulation for a set of specific parameters are shown in *Figure 2*. For these simulations, the possibility of reinitiation was excluded ($t_{\varepsilon > \sigma}=0$). To allow for leaky scanning, the strength of uORF translation initiation was set $t_{\sigma > \varepsilon}=0.8$ (80% of scanning ribosomes convert to elongating ribosomes at the start of the uORF). The rate of elongation in all simulations was modelled as a probability of 0.3 that the ribosome moves during a single tact ($m_\sigma=0.3$). Assuming that an average mammalian ribosome moves five codons per second during elongation (*Ingolia et al., 2011*), the tact of simulation would correspond to 0.06 s (0.3/5). We were unable to find experimental estimates for the velocity of scanning ribosomes in vivo, but in vitro estimates are similar to that of elongating ribosomes (*Vassilenko et al., 2011*; *Berthelot et al., 2004*). Hence, for the simulations shown in *Figure 2*, we used equal rates for elongating and scanning ribosomes ($m_\varepsilon=0.3$). To simulate stress conditions, we tested the model under variable $r_{in}$ from high to absolute zero. It has been estimated that in yeast, on average, the ribosome loads onto mRNA every 0.8 s (*Chu et al., 2014*), which in terms of our model would be a probability of ribosome load of 0.075 per tact. Thus, we decided to model the behavior of the system for $r_{in}$ ranging from 0.1 to 0 (*Figure 2*). In the absence of uORFs, the $r_{out}$ correlates with $r_{in}$ although nonlinearly (yellow curve in *Figure 2A,B*). At the onset of stress, the decrease in $r_{in}$ leads to a disproportionally small decrease in $r_{out}$, but at a near zero value of $r_{in}$, the rates $r_{in}$ and $r_{out}$ begin to decrease proportionally. This is because of changes in the likelihood of ribosome collisions, which reduce their flow. When the load rate is close to zero, very few ribosomes traverse mRNA, making collisions highly unlikely and leading to a direct relationship between $r_{in}$ and $r_{out}$. It appears that the addition of even very short ORFs substantially reduces the number of ribosomes downstream (*Figure 2B*). This is not surprising as in the absence of reinitiation, at least 80% of scanning ribosomes would be

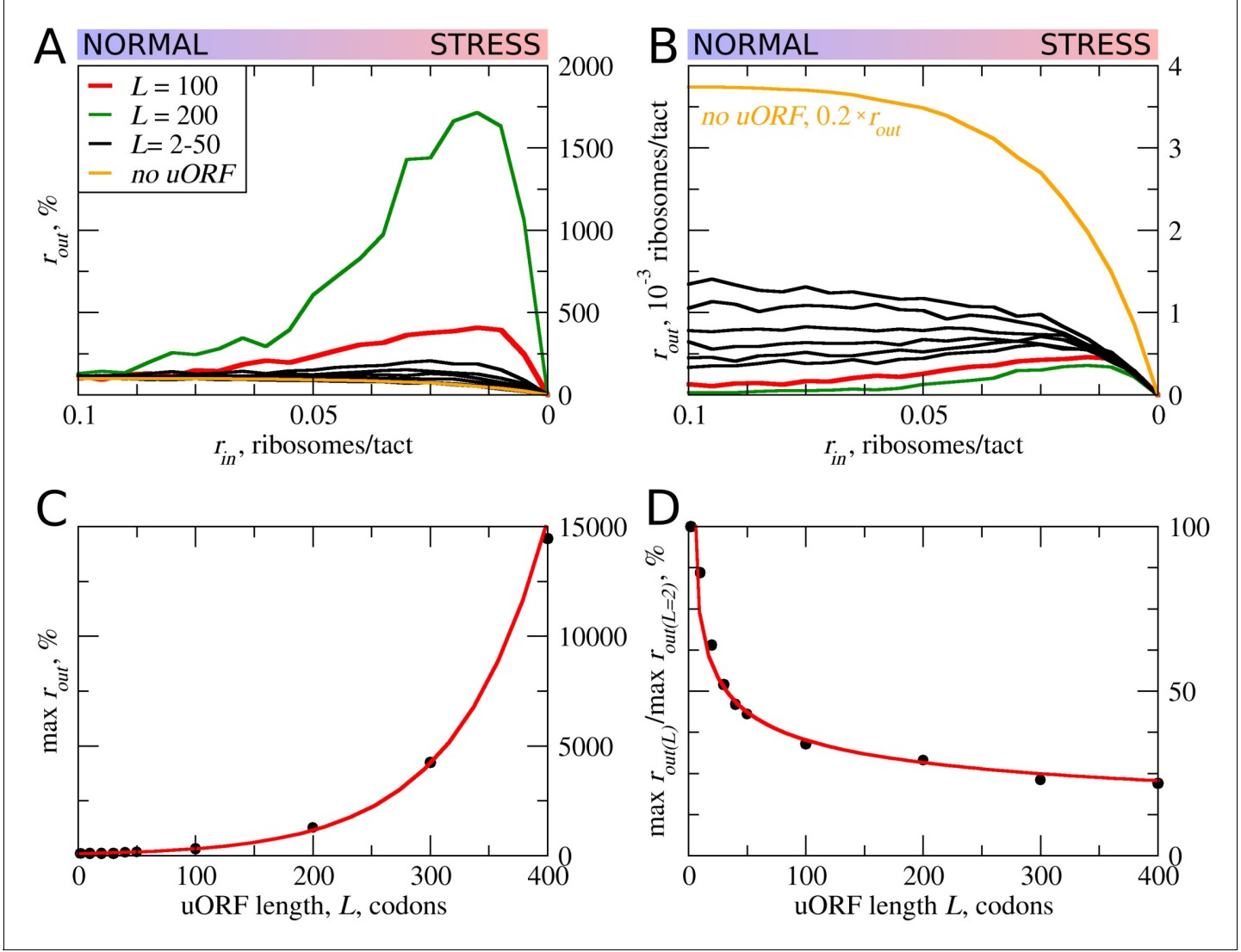

**Figure 2.** uORF length ($L$) influences how the flux of scanning ribosomes changes downstream ($r_{out}$) in response to lowered ribosome load ($r_{in}$) due to stress. Other parameters in the simulations: $t_{\varepsilon > \sigma}$=0; $t_{\sigma > \varepsilon}$=0.8; $m_{\sigma}$=$m_{\varepsilon}$=0.3. (**A, B**) Relative ($r_{out}[r_{in}]/r_{out}[r_{in}$=0.1]) (**A**) and absolute (**B**) changes in the density of ribosomes downstream of a uORF for a range of $r_{in}$ from 0.1 to 0 for mRNAs containing uORFs of different lengths ($L$). (**C**) Dependence of the relative ($r_{out}$) maximum on the uORF length. (**D**) The decrease of maximum $r_{out}$ as a result of increasing uORF length (maximum $r_{out}$ for each $L$ is normalized by $r_{out}$ for $L$= 2). Source data for panel (**A**) are provided in **Figure 2—source data 1** and for panel (**B**) in **Figure 2—source data 2**.
DOI: https://doi.org/10.7554/eLife.32563.003

The following source data is available for figure 2:

**Source data 1.** Source data for **Figure 2A**.
DOI: https://doi.org/10.7554/eLife.32563.004
**Source data 2.** Source data for **Figure 2B**.
DOI: https://doi.org/10.7554/eLife.32563.005

lost. Most interestingly, once the uORF reaches a certain length (between 20 and 30 codons in the simulations shown in **Figure 2**), the relationship between $r_{in}$ and $r_{out}$ becomes non-monotonous and $r_{out}$ starts to increase with decreasing $r_{in}$ at a certain interval. The effect is much more profound for longer uORFs (**Figure 2A,C**). However, this advantage afforded by a uORF in stress resistance comes at a price: as can be seen from **Figure 2B**, uORF-containing mRNAs have lower levels of $r_{out}$ when $r_{in}$ is high and this repression increases with uORF length (**Figure 2D**). This occurs due to

increasing incidence of collisions involving scanning and elongating ribosomes within the uORF and subsequent dissociation of scanning ribosomes.

In other words, according to our ICIER model, long uORFs repress translation of downstream ORFs, which are de-repressed during stress. This is consistent with our earlier study, where we found that the best predictor of stress-resistant mRNAs is the presence of an efficiently translated uORF combined with very low translation of the downstream acORF (*Andreev et al., 2015b*).

## The effect of uORF elongation rate on the flux of scanning ribosomes

In the simulations described above, the movement rates of scanning and elongating ribosomes were set to be equal. However, codon decoding rates can vary significantly depending on their identity and surrounding sequences (for example, see *Shah et al. [2013]*, *Tuller et al. [2010]*, *Wen et al. [2008]*). Thus, the average elongation rate could also vary for different uORFs. This is particularly salient where uORFs encode stalling sites such as in *DDIT3* (*Young et al., 2016*). Therefore, we explored how variations in the rate of ribosome elongation affect the behavior of our model. Even a small decrease in the elongation rate ($m_\varepsilon$) strongly increases the maximum $r_{out}$ relative to its basal level (*Figure 3*). The global (sequence non-specific) rate of elongation may be decreased by the contribution made by stress (for example, by eEF2 phosphorylation) to the stress resistance granted by uORFs. Perhaps more importantly, the local decoding rates may be specifically tuned to the uORF sequences, through nascent-peptide-mediated effects or via RNA secondary structures in 5' leaders.

The stress resistance increases monotonously with decreased elongation rates (*Figure 3C*), but the absolute $r_{out}$ maximum is not monotonous and reaches a maximum when the rate for elongating ribosomes is slightly slower than that for scanning ribosomes (*Figure 3D*). This clearly points to a tradeoff similar to that observed for uORF lengths: the more slowly decoded uORFs provide greater resistance to the stress, but at a cost of greater uORF-mediated repression under normal conditions, thus requiring more mRNA molecules in order to acheive the same protein synthesis rate. However, if the speed of scanning ribosome greatly exceeds that of elongating ribosomes, the increased inhibition under normal conditions does not convert into stress resistance. Relative $r_{out}$ rates, as well as absolute $r_{out}$ rates, drop with increased elongating ribosome velocities once they approach the velocity of scanning ribosomes (*Figure 3A,B*).

It is not clear whether the rates of scanning ribosomes can be influenced locally by nucleotide sequences or whether they can vary under the stress conditions. According to the ICIER modelling, for a maximal performance of uORF-containing mRNAs under stress, there should be an optimal ratio of scanning to elongating ribosomes velocities. Thus, we can hypothesize not only that the codon uORF composition affects local decoding rates but also that the primary sequence of the codon may affect local scanning rates, and may be an important determinant of uORF function.

## Strength and pausing of translation initiation at uORFs and the probability of reinitiation downstream

Translation initiation could often be leaky depending on the context. In such cases, only a proportion of ribosomes would initiate at a start codon, while the remaining ribosomes continue scanning (*Hinnebusch, 2014*; *Kozak, 1999*). This is particularly relevant to uORFs, where even a very weak start codon would yield detectable translation if there were no start codons upstream of it acting to reduce the pool of scanning ribosomes (*Michel et al., 2014a*). Thus, many known regulatory uORFs are initiated at non-AUG starts (*Ivanov et al., 2008, 2018*; *Andreev et al., 2015a*; *Starck et al., 2016*). As expected, increased leakiness of the uORF start (lower $t_{\varepsilon > \sigma}$) elevates the flow of ribosomes downstream of the uORF (*Figure 4A*). The stress resistance is also reduced considerably for uORFs that have leaky starts and disappears when only 30% or fewer initiate at the uORF in the context of other parameters used for the simulations shown in *Figure 4A–D*. This allows us to speculate that uORFs with non-AUG start codons are unlikely to be able to provide stress resistance, as such initiation codons are non-optimal by default and should be prone to leaky scanning.

Our default model does not take into account that initiation is a slow process. It is conceivable that the time that the initiating ribosome spends on the start codon may affect the stress resistance and it is likely that this initiation dwell time may vary for different start codons. We therefore carried out simulations in which elongating ribosomes were paused at the start immediately after their conversion from scanning ribosomes. The length of delay was set as a probability ($\delta_\varepsilon$) that determines

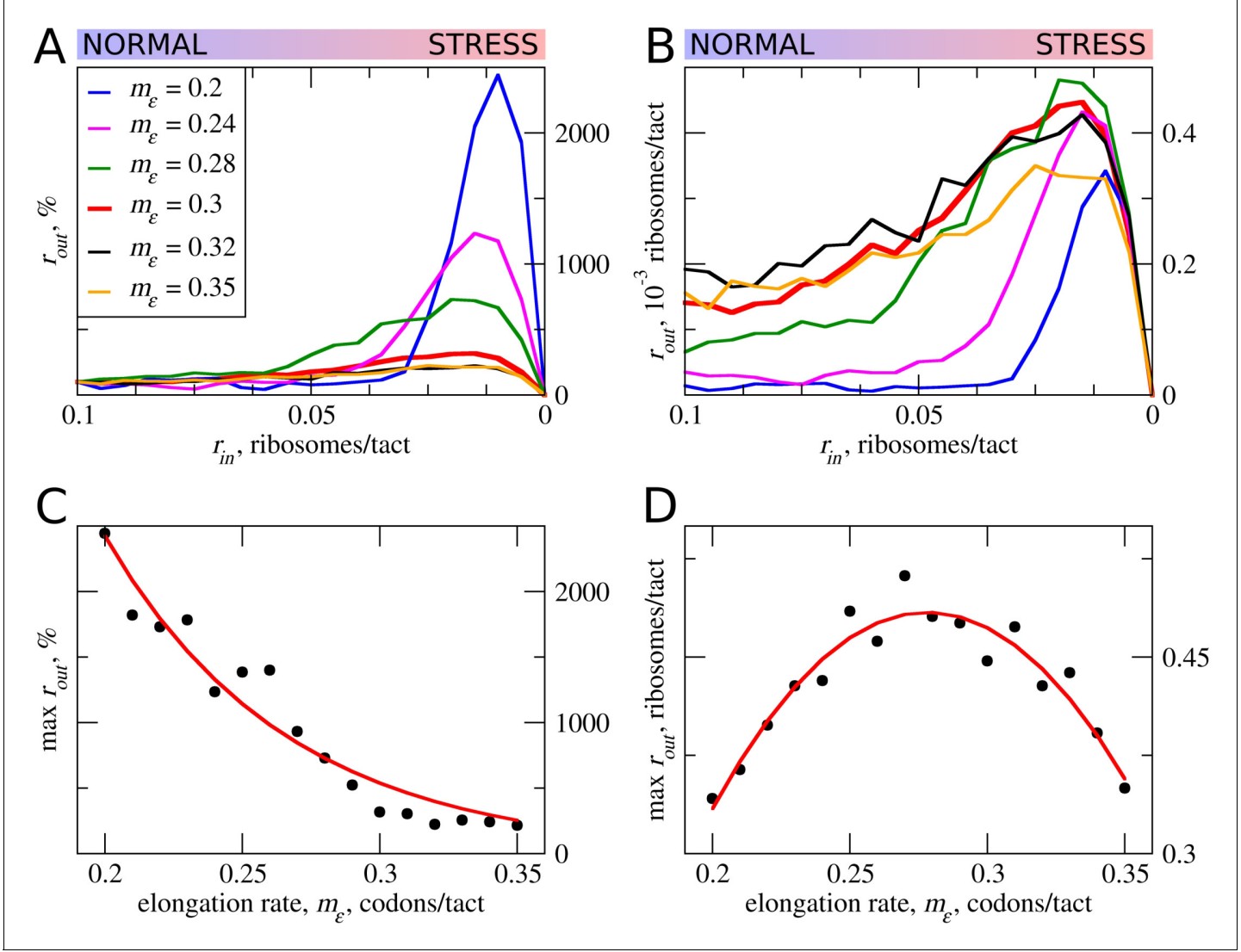

**Figure 3.** The effect the movement rates ($m_e$) for elongating ribosomes on stress resistance. Other parameters of the simulations $t_{\varepsilon>\sigma}$=0; $t_{\sigma>\varepsilon}$=0.8; $L$=100; $m_\sigma$=0.3. (A, B) Relative ($r_{out}[r_{in}]/r_{out}[r_{in}=0.1]$) (A) and absolute (B) changes in the density of ribosomes downstream of the uORF for a range of $r_{in}$ from 0.1 to 0 for different elongating ribosome movement rates ($m_\varepsilon$)). (C) Dependence of the relative ($r_{out}$) maximum increase on elongating ribosome movement $m_\varepsilon$. (D) Dependence of the absolute maximum $r_{out}$ rate on the $m_\varepsilon$. Source data for panel (A )are provided in **Figure 3—source data 1** and for panel (B) in **Figure 3—source data 2**.

DOI: https://doi.org/10.7554/eLife.32563.006

The following source data is available for figure 3:

**Source data 1.** Source data for **Figure 3A**.
DOI: https://doi.org/10.7554/eLife.32563.007
**Source data 2.** Source data for **Figure 3B**.
DOI: https://doi.org/10.7554/eLife.32563.008

the ability of paused elongating ribosomes to move (**Figure 4E,F**). Apparently, the increased delay reduces the inhibitory effect of uORFs and reduces the stress resistance. This is an expected result in the context of the model, as the delay of initiating ribosomes increases their distance to the downstream scanning ribosomes and thus reduces the chance of the collision.

In all of the above simulations, reinitiation of ribosomes terminating at the uORF was disallowed, that is $t_{\varepsilon>\sigma}$=0. In practice, however, a significant proportion of ribosomes can reinitiate downstream depending on specific uORF features (**Gunišová et al., 2016**; **Mohammad et al., 2017**;

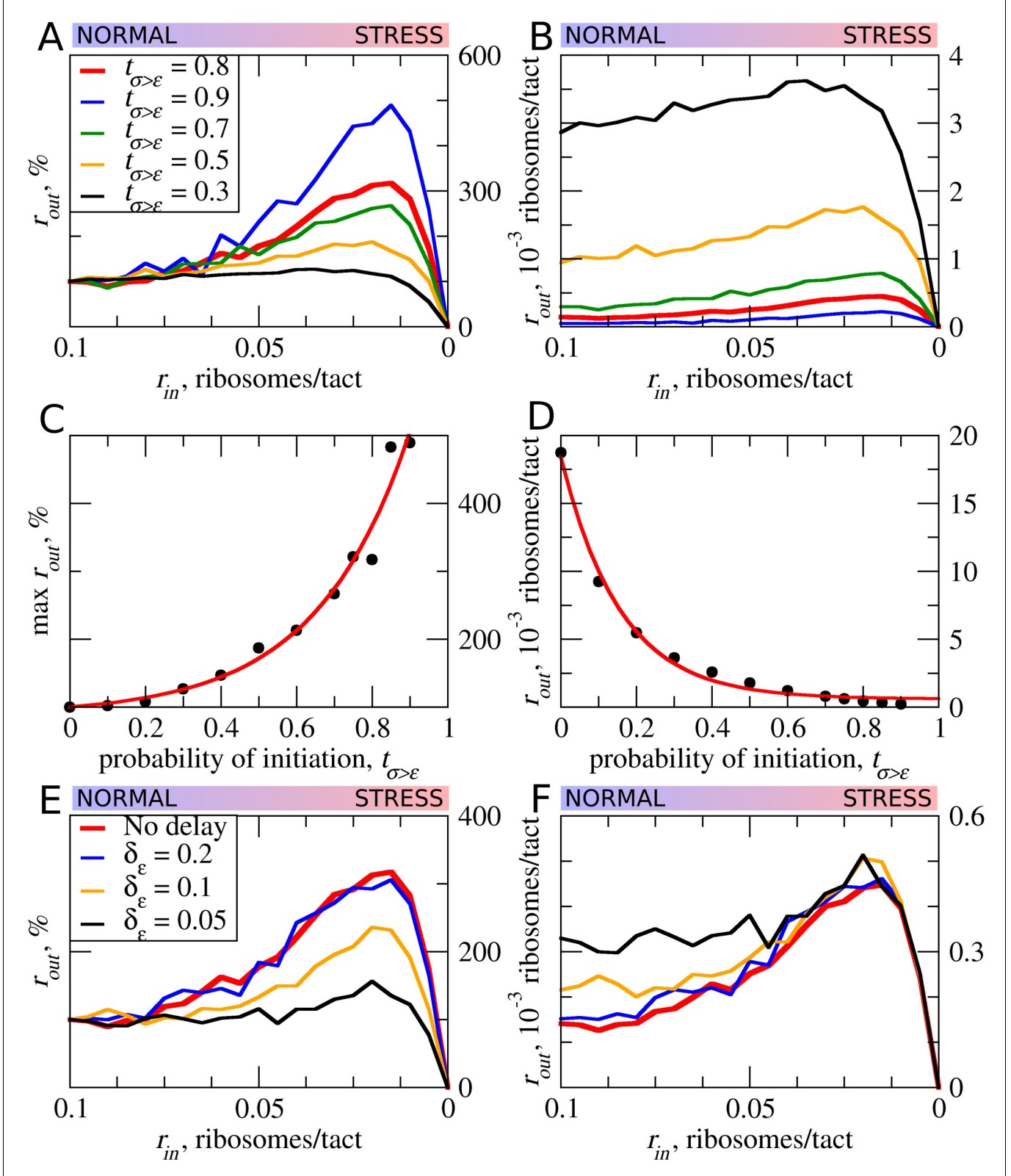

**Figure 4.** The effect of uORF initiation efficiency ($t_{\sigma>\varepsilon}$) on stress resistance. Other parameters of the simulations $t_{\varepsilon>\sigma}$=0; $L$=100; $m_\sigma$=0.3; $m_\varepsilon$=0.3. (**A, B**) Relative ($r_{out}[r_{in}]/r_{out}[r_{in}$=0.1]) (**A**) and absolute (**B**) changes in the density of ribosomes downstream of a uORF for a range of $r_{in}$ from 0.1 to 0 for different uORF initiation efficiencies ($t_{\sigma>\varepsilon}$). (**C**) Dependence of the relative ($r_{out}$) maximum increase on the uORF initiation efficiency $t_{\sigma>\varepsilon}$. (**D**) Dependence of absolute maximum $r_{out}$ rate on $t_{\sigma>\varepsilon}$. (**E, F**) Relative ($r_{out}[r_{in}]/r_{out}[r_{in}$=0.1]) (**E**) and absolute (**F**) changes in the density of ribosomes downstream of a uORF

*Figure 4 continued on next page*

*Figure 4 continued*

depending on the delay $\delta_\varepsilon$, which is the probability that newly formed elongating will move. For no delay, $\delta_\varepsilon$=1. Source data for panel (A) are provided in *Figure 4—source data 1*, for panel (B) in *Figure 4—source data 2*, for panel (E) in *Figure 4—source data 3* and for panel (F) in *Figure 4—source data 4*.

DOI: https://doi.org/10.7554/eLife.32563.009

The following source data is available for figure 4:

**Source data 1.** Source data for *Figure 4A*.
DOI: https://doi.org/10.7554/eLife.32563.010
**Source data 2.** Source data for *Figure 4B*.
DOI: https://doi.org/10.7554/eLife.32563.011
**Source data 3.** Source data for *Figure 4E*.
DOI: https://doi.org/10.7554/eLife.32563.012
**Source data 4.** Source data for *Figure 4F*.
DOI: https://doi.org/10.7554/eLife.32563.013

*Young et al., 2015*). Intuitively, uORFs that allow reinitiation downstream should be less inhibitory, for example, when all ribosomes reinitiate downstream of the uORF with an optimal non-leaky start, all ribosomes that engage with mRNA would also translate acORF. It is therefore also likely that reinitiation would reduce the ability of a uORF to provide stress resistance. We explored how reinitiation affects stress resistance in the context of the ICIER model (*Figure 5*) by permitting terminating ribosomes to convert to scanning ribosomes with a certain probability. As expected, elevated reinitiation reduces the inhibitory effect of uORFs as well as the stress-protective effect of uORFs. It appears that a dramatic effect could be achieved even at very low reinitiation rates. In the simulations shown in *Figure 5*, the significant reduction in stress resistance can be observed with a reinitiation rate as low as 1% and the resistance almost disappears when just 5% of the ribosomes reinitiate downstream. On the basis of these observations, we predict that single uORFs that enable stress resistance do not allow reinitiation to take place, contrasting with the case of mRNAs such as *GCN4* that have a combination of uORFs and achieve a protective effect through delayed reinitiation (*Hinnebusch, 1993*).

## Miscellaneous parameters of the model: ribosome gabarits and fall off rates

In all of the simulations so far, the only condition considered in which scanning ribosomes become dissociated from mRNA was when colliding with upstream elongating ribosomes. However, the complexes of scanning ribosomes with mRNA are unstable because their co-isolation with mRNA requires chemical cross-linking (*Archer et al., 2016*; *Valásek et al., 2007*). It is therefore likely that scanning ribosomes dissociate from mRNA even in the absence of collisions with other ribosomes. The factors that affect the stability of scanning ribosomes in complex with mRNA are unclear, for example, it does not seem that the length of the 5' leaders has a substantial effect on the probability that scanning ribosomes will dissociate from mRNA (*Vassilenko et al., 2011*; *Dmitriev et al., 2007*). Without adequate knowledge of the factors influencing the dissociation of scanning ribosomes from mRNA, we decided to model that process as spontaneous dissociation from mRNA using varying dissociation probabilities. First, we explored how such spontaneous dissociation affects our default model when scanning ribosomes are being removed by upstream elongating ribosomes (*Figure 6A*). It appears that such spontaneous dissociation does not significantly affect the inhibitory effect of uORFs, but they do reduce the stress-protective properties of those uORFs.

Although elongating ribosome drop-off is very low in mammals (*Guo et al., 2014*) and even in bacteria (*Oh et al., 2011*; *Sin et al., 2016*), we explored how different rates of drop-off could affect the behavior of our model. It appears that increased drop-off reduces both the inhibitory and the stress-protective properties of uORFs (*Figure 6B*).

Further, we explored how the system would behave if scanning ribosomes only dissociated spontaneously, that is, when collisions do not cause dissociations (*Figure 6C*). In this case, uORFs lose any stress-protective properties.

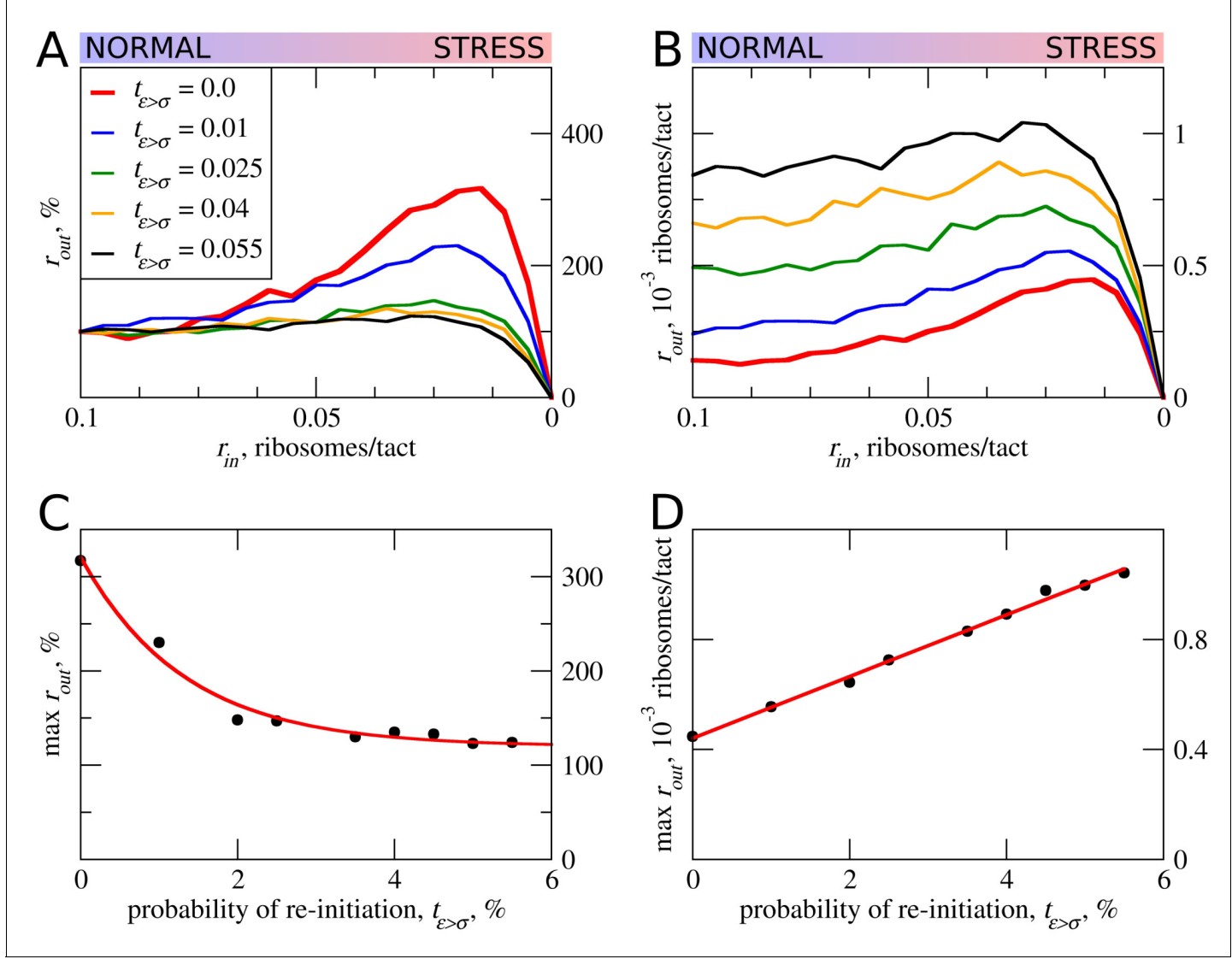

**Figure 5.** The effect of reinitiation downstream of the uORF on stress resistance. Other parameters of the simulations $t_{\varepsilon>\sigma}=0$; $t_{\sigma>\varepsilon}=0.8$; $L=100$; $m_{\sigma}=0.3$; $m_{\varepsilon}=0.3$. (**A, B**) Relative ($r_{out}[r_{in}]/r_{out}[r_{in}=0.1]$) (**A**) and absolute (**B**) changes in the density of ribosomes downstream of a uORF for a range of reinitiation probabilities ($t_{\varepsilon>\sigma}$ from 0 to 0.055). (**C**) Dependence of the relative ($r_{out}$) maximum increase on the reinitiation probability $t_{\varepsilon>\sigma}$. (**D**) Dependence of absolute maximum $r_{out}$ rate on the $t_{\varepsilon>\sigma}$. Source data for panel (**A**) are provided in *Figure 5—source data 1* and for panel (**B**) in *Figure 5—source data 2*.

DOI: https://doi.org/10.7554/eLife.32563.014

The following source data is available for figure 5:

**Source data 1.** Source data for *Figure 5A*.
DOI: https://doi.org/10.7554/eLife.32563.015
**Source data 2.** Source data for *Figure 5B*.
DOI: https://doi.org/10.7554/eLife.32563.016

Ribosome structures are dynamic and they undergo conformational changes as they move along mRNA, for example, they protect mRNA fragments of different size depending on whether the POST- or PRE-translocation complex is being stabilized (*Lareau et al., 2014*). Scanning ribosomes also change the length of protected fragments depending on their conformations and on the influence of the translation initiation factors with which they are associated (*Archer et al., 2016*; *Pisarev et al., 2008*). To explore how the length of the space that ribosomes occupy on mRNA influences our model, we carried out a series of simulations in which the length of elongating ribosomes

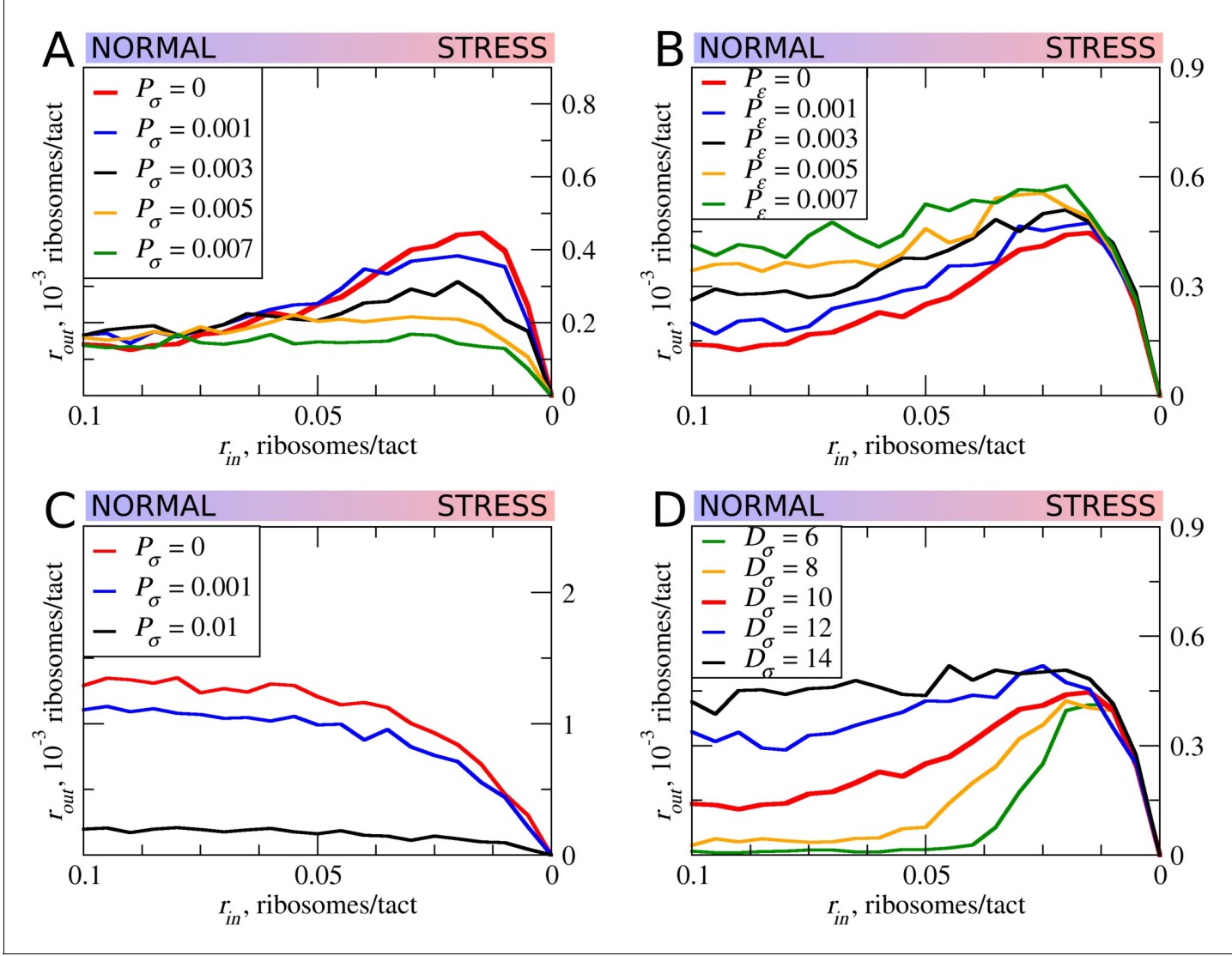

**Figure 6.** The effect of the size of scanning ribosomes ($D_\sigma$) and probability of spontaneous ribosome dissociation ($P_\sigma$) on stress resistance. Other parameters of the simulations; $t_{\sigma>\varepsilon}$=0.8; $L$=100; $m_\sigma$=0.3; $m_\varepsilon$=0.3. (A, B) Absolute changes in the density of ribosomes downstream of a uORF under a scenario in which scanning (A) or elongating (B) ribosomes could dissociate spontaneously with probabilities $P_\sigma$ and $P_\varepsilon$, respectively. (C) Same as (B) but scanning ribosomes do not dissociate upon collisions with elongating ribosomes. (D) The absolute changes in the densities of ribosomes downstream of a uORF depending on the size of the scanning ribosomes. Source data for panel (A) are provided in *Figure 6—source data 1*, for panel (B) in *Figure 6—source data 2*, for panel (C) in *Figure 6—source data 3* and for panel (D) in *Figure 6—source data 4*.
DOI: https://doi.org/10.7554/eLife.32563.017

The following source data is available for figure 6:

**Source data 1.** Source data for *Figure 6A*.
DOI: https://doi.org/10.7554/eLife.32563.018
**Source data 2.** Source data for *Figure 6B*.
DOI: https://doi.org/10.7554/eLife.32563.019
**Source data 3.** Source data for *Figure 6C*.
DOI: https://doi.org/10.7554/eLife.32563.020
**Source data 4.** Source data for *Figure 6D*.
DOI: https://doi.org/10.7554/eLife.32563.021

was kept constant at 10 codons, while the length of scanning ribosomes was varied (*Figure 6D*). Counterintuitively, we observed substantial differences. Both the inhibitory effect of uORFs and their stress-protective properties appear to be much greater in the case of 'short' scanning ribosomes. Without knowledge of mechanistic aspects of ribosome collisions and their consequences, this observation gives us little insight, except for the fact that the dimensions of scanning and elongation ribosomes are important parameters that can influence uORF performance under stress conditions.

## Correlations with existing ribosome-profiling data

While our model cannot make accurate predictions of stress-resistance levels for specific mRNAs, it makes certain predictions regarding the general features of uORFs that are associated with the stress-protective properties of uORFs occurring in mRNA 5′ leaders not containing other uORFs. Specifically, according to our model, stress protective uORFs should not permit significant reinitiation downstream. Longer uORFs are expected to be more protective than short uORFs, and those that are initiated more efficiently should be more stress protective than those that are poorly translated. In addition, uORFs containing slowly decoded regions should also be stress-protective. To explore whether these predictions hold true, we focused on the data obtained in three independents studies carried out in HEK293T cells where an ISR was instigated with either arsenite (*Andreev et al., 2015b*; *Oh et al., 2016*) or tunicamycin treatments (*Sidrauski et al., 2015*).

For these purposes, we selected 325 transcripts containing single uORFs whose translation is supported by aggregated ribosome profiling data from studies that are currently publicly available in the GWIPS-viz browser (*Michel et al., 2014b*, *2018*). We then correlated the features of these uORFs with the stress resistance of their corresponding mRNAs, which was estimated as a Z-score upon Z-score transformation as in our earlier studies (*Andreev et al., 2015a*, *2015b*) (see 'Materials and methods'). The measurement of uORF length is straightforward, it was calculated as the number of codons between the first ATG codon of a uORF and its stop codon. Indeed, as predicted by the model, we found strong and statistically significant positive correlation between the length of a uORF and the stress-resistance of its corresponding mRNA in all three studies (*Figure 7A*). To estimate the efficiency of translation of uORFs, we measured their ribosome occupancy under control conditions (see 'Materials and methods'). This measurement was made under the assumption that the translation initiation efficiency of a uORF is a major factor in determining the average density of footprints aligning to that uORF. We found positive and statistically significant correlation between uORF footprint density and stress-protective properties only in one of the three studies (*Figure 7B*). To estimate the strength of ribosome pauses in uORFs, we compared the magnitude of the highest peak of density in the uORF relative to its average footprint density using transcriptome alignments of the data from the studies available in the GWIPS-viz (*Michel et al., 2018*) (see Materials and methods). Although we observed positive correlations between pause scores and stress resistance in all three studies, they appeared weak and not statistically significant (*Figure 7C*).

There are several possible explanations for the lack of statistically significant correlations between average footprint densities, pause scores and stress resistance that do not necessarily invalidate the predictions of our model. First, it is expected that our estimates of initiation and pausing, which are based on ribosome profiling data, are insufficiently accurate because of the short lengths of the uORFs and as the result of a mappability and sequencing biases (*O'Connor et al., 2016*). Second, the selection of mRNA with single uORFs necessitated the choice of uORFs that fit only a certain range of property parameters, for example if a uORF's translation efficiency is low, we may not be able to detect such a uORF as being translated. Finally, the distribution of uORFs with different features is intrinsically non-random as a result of evolutionary selection imposed by functional properties of uORFs. Nevertheless, the observation that uORF length is important and is a significant determinant of stress resistance supports the results of our ICIER modelling. In all three studies, uORF containing mRNAs with high Z-score (>2) are completely depleted of very short uORFs which, under our simulations, are not able to provide stress resistance because they are affected by ribosome collisions.

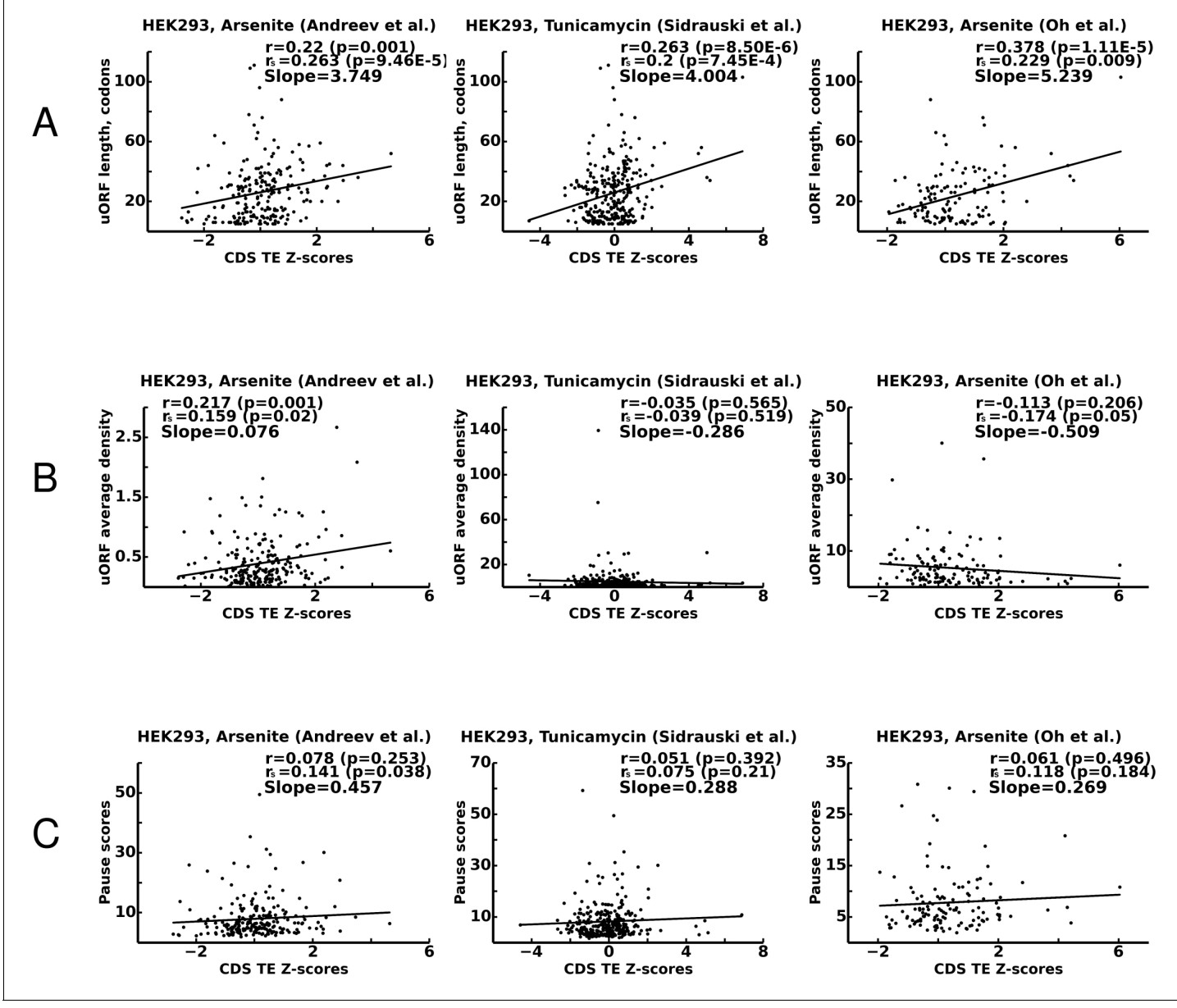

**Figure 7.** Correlations between translational stress resistance of mRNAs and features in the 5' leaders of uOFRs for three studies. (**A**) uORF length. (**B**) uORF translation initiation measured as uORF translation efficiency. (**C**) Ribosome pausing estimated with pause score. Pearson correlation coefficients (r) and Spearman rank correlations ($r_s$) are provided for each plot along with the corresponding p-values. The slope of linear regression is also indicated for each plot. Source data are provided in *Figure 7—source data 1*.

DOI: https://doi.org/10.7554/eLife.32563.022

The following source data is available for figure 7:

**Source data 1.**

DOI: https://doi.org/10.7554/eLife.32563.023

## Discussion

Using the ICIER model developed in this work, we explored how the flow of scanning ribosomes across an mRNA leader is changed in the presence of a translated ORF depending on several parameters such as ORF length, initiation efficiency, decoding rate and reinitiation probability. Our results demonstrate that for mRNAs with certain uORFs, the relationship is not monotonous. It appears that within a certain range, the rate of PIC assembly has an inverse relationship with ribosome availability

downstream of uORFs. We explored how several different parameters of uORFs affect this phenomenon. It appears that the non-monotonous behavior emerges only after uORFs reach a certain length. The elevation of ribosome flow downstream of a uORF is higher for longer uORFs. However, uORF inhibitory effects on downstream translation also increase with their length. Thus, there is a clear trade-off: longer uORFs provide stronger resistance to the stress, but the translation efficiency under normal conditions is lower for ORFs downstream of longer uORFs. Consequently, to achieve the same protein synthesis rate, as the length of the uORF increases, more mRNA copies need to be synthesized. This trade-off probably shapes the specific lengths of regulatory uORFs and the steady-state levels of the corresponding mRNAs.

Decreased elongation rates increase the stress resistance. A global decrease in elongation rates is expected under certain stress conditions, for example, during stress resulting from eEF2K-mediated phosphorylation of eEF2 (*Leprivier et al., 2013*) or due to decreased concentration of available aminoacylated tRNAs. However, the decoding rates may vary among different uORFs as specific sequences affect the speed of elongation. Indeed, it seems that at least some stress-resistant mRNAs contain stalling sites (*Young et al., 2016*). It should be noted that slow elongation at the uORF also incurs a cost, there is a tradeoff between absolute levels of translation under stress and slow elongation. Interestingly, the optimal performance of a uORF as a stress resistor seems to be achieved at a certain ratio of scanning and elongation ribosome velocities.

In all of the simulations described above, irrespective of the specific parameters used, the stress resistance (when observed) occurred within a specific narrow range of scanning-ribosome loading on mRNA ($r_{in}$ ~0.02). It is reasonable to expect that the stress resistance of specific mRNAs would be tuned to particular loading levels. We equated stress levels to $r_{in}$ rates, but it is likely that different mRNAs have different $r_{in}$ rates under the same cellular conditions. The exact sequence at the 5' end of the mRNA could potentially influence eIF4F assembly (*Zinshteyn et al., 2017*) and subsequently the rate with which the PIC is assembled on the mRNA. Indeed, the effect of the mRNA 5' end on single-uORF-dependent stress resistance was recently demonstrated (*Andreev et al., 2015b*): the addition of a strong hairpin to the 5' end of the *IFRD1* leader alleviated stress resistance, even though the uORF itself remained intact. Also, according to our simulations, leaky initiation at the uORF begins to reduce the stress resistance, as does the possibility of re-initiation downstream of the uORF. This highlights the importance of uORF flanking sequences (upstream for modulating PIC recruitment [$R_{in}$] and initiation efficiency, and downstream for reinitiation) on uORF performance. Thus, despite the central role of uORFs, other sequences of the 5' leader can also significantly influence mRNA translation under stress.

To validate our model, we explored how uORF features correlate with the stress resistance of mRNAs using ribosome-profiling data obtained under stress conditions in studies from three independent laboratories (*Andreev et al., 2015b*; *Sidrauski et al., 2015*; *Oh et al., 2016*). We found that uORF length strongly correlates with their stress-protective properties. The model predicts that the efficiency of translation initiation should also correlate with the stress-protective properties of uORFs, but we found support for this prediction in only one of the three studies (*Andreev et al., 2015b*). Only weak and statistically insignificant correlations were found between the strength of ribosome pauses in uORFs and stress resistance. Besides technical difficulties in estimating translation initiation efficiency and elongation rates from ribosome-profiling data, the relationship of the last two parameters and stress resistance may be convoluted by non-random features of naturally occurring uORFs. A more 'uORF centered' experimental approach is required to explore the details uORF functions under stress, for example, with ribosome profiling under conditions of various degree of ISR (PIC depletion), with precise mapping of 5' leaders (which can be achieved with nano CAGE) to uncover more mRNA species with a single uORF, and with very deep coverage for more accurate uORF riboseq signal quantification.

In conclusion, by using a simple model of scanning and elongating ribosome interference, we demonstrated that a single uORF can be sufficient to provide mRNA translation control with resistance to TC depletion which occurs during ISR. The general principle of uORF-mediated stress resistance seems to be based on strong repression of downstream translation under normal conditions, which is derepressed during the stress. The exact parameters of stress protective uORFs are thus a product of the trade-off between the inhibition of absolute levels of translation and its relative increase during the stress. The uORFs providing the resistance are likely to be longer, their initiation sites are expected to have a low level of leakiness (but some level of leakiness is essential), they may

contain slowly decoded sites and they should not permit high levels of reinitiation downstream. Given that a single uORF could be such a versatile regulatory element, a combination of regulatory uORFs could generate very complex behaviors. Many eukaryotic mRNAs contain multiple uORFs and their concerted impact should be the subject of future experimental and theoretical studies. Obtaining accurate values for ICIER modelling and further improvements of the model may eventually aid the design of mRNA leaders that have a desired response to stress.

# Materials and methods

## ICIER implementation and computer simulations

The mRNA was modeled by a string of $L = L_1 + L_2 + L_3$ symbols 0, 1, 2 with uORF modeled by the $L_2$ symbols in the middle part of the string. Symbol 0 corresponds to an unoccupied site (codon). A block of $D_\sigma$ consecutive symbols 1 corresponds to $D_\sigma$ sites occupied by a scanning ribosome. A block of $D_\varepsilon$ consecutive symbols 2 corresponds to $D_\varepsilon$ sites occupied by an elongating ribosome.

At every tact (time step), the code implements several routines modeling the random processes described in the subsection 'The model of initiation complexes interference with elongating ribosomes (ICIER)': attachment of scanning ribosomes to the mRNA, ribosomes movement, dissociation of ribosomes from the mRNA, initiation of scanning ribosomes and reinitiation of elongating ribosomes. More precisely, one time-step of the algorithm includes the following routines:

1. If there are at least $D_\sigma$ unoccupied sites at the beginning of the mRNA, then the algorithm adds a scanning ribosome to the first $D_\sigma$ sites with probability λ (by changing the first $D\sigma$ symbol 0 to 1).
2. If the end symbol 1 of a scanning ribosome is at the initial site of uORF and (in case $D_\varepsilon > D_\sigma$) there are enough unoccupied sites to fit an elongating ribosome, then this scanning ribosome is replaced with an elongating ribosome with probability $P_{\sigma>\varepsilon}$.
3. If the end symbol 2 of an elongating ribosome is at the last site of the uORF and (in case $D_\varepsilon < D_\sigma$) there are enough unoccupied sites to obtain a scanning ribosome, then the elongating ribosome is replaced with a scanning ribosome with probability $P_{\varepsilon>\sigma}$ and is removed with probability $1-P_{\varepsilon>\sigma}$.
4. A scanning ribosome reaching the last site of mRNA is removed from the mRNA (by changing 1s to 0s), and the algorithm registers this ribosome as reaching the final position.
5. At every other site of the mRNA, every ribosome either moves or stays according to the following rules:
   1. A scanning ribosome moves one step forward with probability $m_\sigma$ if there is an empty site in front of it.
   2. An elongating ribosome moves one step forward with probability $m_\varepsilon$ if the site in front of it is not occupied by an elongating ribosome. If the site in front of a moving elongating ribosome is 1, meaning that there is a scanning ribosome in front of it, then this scanning ribosome is removed from the mRNA.
6. Each scanning and elongating ribosome is removed from the mRNA with probability $P_\sigma$ and $P_\varepsilon$, respectively.

The algorithm was implemented in the MATLAB software package. The simulations were performed on a standard MacBook computer.

Each data point for *Figures 1–6* was generated by running the code for $10^6$ time steps. A typical curve in these figures contains 20 data points corresponding to different values of the loading probability λ ($r_{in}$).

The computer code is freely available at GitHub: https://github.com/maximarnold/uORF_TASEP_ICIER (*Arnold, 2018*; copy archived at https://github.com/elifesciences-publications/uORF_TASEP_ICIER).

Parameters of all simulations described in this manuscript and their outcomes are provided in *Supplementary file 1*.

## Analysis of ribosome-profiling data

For the detection of reliably translated uORFs, data aggregated from a subset of the studies currently available in the GWIPS-viz browser (*Michel et al., 2018*) were used. Specifically, the data were

pulled from the following studies: *Andreev et al. (2015b)*, *Sidrauski et al. (2015)*, *Guo et al. (2014)*, *Oh et al. (2016)*, *Park et al. (2017)*, *Lintner et al. (2017)*, *Park et al. (2016)*, *Xu et al. (2016)*, *Calviello et al. (2016)*, *Tirosh et al. (2015)* and *Werner et al. (2015)*. APPRIS principal iso-forms (*Rodriguez et al., 2013*) with annotations from Gencode version 25 were used. All ORFs located within the 5' leader of Gencode transcripts not overlapping CDS were ranked based on their translational efficiency. The top 5% were manually examined for the consistency of ribo-seq signal and the absence of other translated regions in the 5' leaders. mRNAs with more than one uORF were excluded from the analysis. In total, 325 transcripts were accepted as transcripts with a single uORF. Gencode transcript IDs and the coordinates of uORFs are provided in *Supplementary file 2*.

To score mRNAs as stress-resistant, we used data from three independent studies in which ISR was triggered in HEK293T cells either with arsenite (*Andreev et al., 2015b*; *Oh et al., 2016*) and tunicamycine (*Sidrauski et al., 2015*). Translation efficiency for all transcripts was calculated using control and treatment conditions (data obtained in the *DDX3* knockout cell line from *Oh et al. (2016)* were excluded) by normalizing the number of ribosome footprints aligning to CDS region over the average RNA-seq density and CDS length. Differential translation was measured upon Z-score transformation as in our earlier studies (*Andreev et al., 2015a*, *2015b*). Specifically transcripts were grouped into bins of 300 based on the similarity of their expression levels, and Z-scores were computed for each transcript based on log-fold changes of their translational effi-ciency between two conditions. These Z-scores were used as a measure of translational response to stress. The correlation was measured between Z-scores and each of the three features of uORFs: length, translation efficiency and pause scores (shown in *Figure 7*). Pause scores were calculated using the same pooled data as those used for the detection of uORFs (*Andreev et al., 2015b*; *Sidrauski et al., 2015*; *Guo et al., 2014*; *Oh et al., 2016*; *Park et al., 2017*; *Lintner et al., 2017*; *Park et al., 2016*; *Xu et al., 2016*; *Calviello et al., 2016*; *Tirosh et al., 2015*; *Werner et al., 2015*) by dividing the highest ribosome density at a single uORF location by the average ribosome foot-print density of the corresponding uORF (excluding the highest peak coordinate).

## Acknowledgements

This work was supported by Science Foundation Ireland grant (12/IA/1335) to PVB and Russian Sci-ence Foundation grant (RSF16-14-10065) to DEA. DR acknowledges the support of the National Sci-ence Foundation (NSF) through grant DMS-1413223. SJK and AMM wishes to acknowledge individual support from the Irish Research Council.

## Additional information

### Funding

| Funder | Grant reference number | Author |
| --- | --- | --- |
| Science Foundation Ireland | 12/IA/1335 | Pavel V Baranov |
| National Science Foundation | DMS-1413223 | Dmitry Rachinskiy |
| Russian Science Foundation | RSF16-14-10065 | Dmitry E Andreev |
| Irish Research Council | | Stephen J Kiniry<br>Audrey M Michel |

The funders had no role in study design, data collection and interpretation, or the decision to submit the work for publication.

### Author contributions

Dmitry E Andreev, Conceptualization, Funding acquisition, Investigation, Methodology, Writing—original draft, Writing—review and editing; Maxim Arnold, Conceptualization, Software, Formal anal-ysis, Investigation, Methodology, Writing—original draft, Writing—review and editing; Stephen J Kiniry, Investigation, Visualization, Methodology, Writing—review and editing; Gary Loughran, Inves-tigation, Writing—review and editing; Audrey M Michel, Supervision, Investigation, Writing—review and editing; Dmitrii Rachinskii, Conceptualization, Software, Supervision, Funding acquisition,

Investigation, Methodology, Writing—original draft, Writing—review and editing; Pavel V Baranov, Conceptualization, Formal analysis, Supervision, Funding acquisition, Investigation, Visualization, Methodology, Writing—original draft, Writing—review and editing

### Author ORCIDs
Gary Loughran http://orcid.org/0000-0002-2683-5597
Pavel V Baranov http://orcid.org/0000-0001-9017-0270

### Decision letter and Author response
Decision letter https://doi.org/10.7554/eLife.32563.029
Author response https://doi.org/10.7554/eLife.32563.030

## Additional files

### Supplementary files
• Supplementary file 1. The parameters of the computer simulations.
DOI: https://doi.org/10.7554/eLife.32563.024

• Supplementary file 2. The coordinates of the uORFs.
DOI: https://doi.org/10.7554/eLife.32563.025

• Transparent reporting form
DOI: https://doi.org/10.7554/eLife.32563.026

### Data availability
All data generated during this study are included in the manuscript and supporting files. Source data files have been provided for Figures 2 to 7.

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
