## [Decision Letter]

Thank you for submitting your article "TASEP modelling provides a parsimonious explanation for the ability of a single uORF to upregulate downstream ORF translation during the Integrated Stress Response" for consideration by *eLife*. Your article has been reviewed by three peer reviewers, including Nahum Sonenberg as the Reviewing Editor and Reviewer #1, and the evaluation has been overseen by James Manley as the Senior Editor.

The reviewers have discussed the reviews with one another and the Reviewing Editor has drafted this letter to help you assess the significant concerns of the reviewers and the feasibility of a response that requires further analysis and possibly experimental support for your computational model. Please respond with an action plan and time table for the completion of the essential additional work. The Board and reviewers with then evaluate your response and come back with further advice.

Summary:

The in-silico approach in the current study explains features of the uORF that favor downstream initiation despite a reduction in ribosome input. The manuscript addresses a timely and important question in the field: what are the mechanisms by which a single uORF can trigger preferential translation in the ISR. The presented models begin to lay out themes that can be applied genome-wide.

Essential revisions:

This is a computational study that follows their previous publication in *eLife*.

As stated by one of the reviewers (see below), some of the underlying assumptions to the computation model are not as well supported in the literature. While key themes are not tested at least some correlations in emerging seq- databases should be provided. Please respond to the constructive and thoughtful criticisms of the reviewers as described below.

Reviewer #1:

Andreev et al. use a computational model to examine the effect of various features of a single uORF on downstream scanning ribosomes. This study is a follow-up of a previous work by the same group (Andreev, et al., 2015), which demonstrated that several human mRNAs need only one uORF to render them resistant to Integrated Stress Response (ISR)-mediated translation inhibition. This constitutes a significant advance in the field. The in-silico approach in the current study explains features of the uORF that favor downstream initiation despite a reduction in ribosome input. As a computational/bioinformatics paper, the analysis provides important insight into the understanding of how mRNAs with only one uORF are resistant to ISR-mediated translation inhibition.

Reviewer #2:

This manuscript describes a modeling platform designated ICIER, which is based on TASEP, to explain how single uORFs can differentially regulate translation at downstream coding sequences during the Integrated stress response (ISR). In the ISR, it is known that eIF2 phosphorylation during stress depletes the eIF2 ternary complex, reducing global translation initiation. Accompanying this global translation reduction, some mRNAs remain efficiently translated (termed resistant or tolerant), while others are actually preferentially translated. There are several models to explain the mechanism of preferential translation, with the core models featuring critical functions for uORFs. Several ISR genes, including DDIT3/CHOP and GADD34/PPPR15B feature a single uORF that is sufficient for preferential translation and it is proposed that the start codon context and delayed elongation are two critical features for their preferential translation during eIF2 phosphorylation.

The premise of the manuscript is that scanning 40S ribosomes dissociate from the 5'-leader of mRNAs when they proceed into (collide) with elongating ribosomes that are translating an uORF. In a sense, scanning ribosomal units can queue up behind an elongating ribosome. The interface between scanning and elongating ribosomes reflects the amount of the preinitiation ribosomal complex loading onto the 5'-end of the mRNAs and the efficiency of elongating ribosomes. In the case of *DDIT3* there is a reported ribosomal stall, which could exacerbate the queue of scanning ribosomes. Other mechanisms, some linked to stress, may also lower translation elongation of the uORFs. The manuscript modeling suggests that the uORF length can change the likelihood of ribosomal collisions between scanning ribosomes and those elongating. Furthermore, eIF2 phosphorylation may alter the loading of ribosomes onto mRNAs and hence the flux of scanning ribosomes that can queue up. The manuscript modeling suggests that resistant mRNA translation of the coding sequence is enhanced when elongating ribosomes at uORFs are slight slower that the incoming scanning ones. The slower the uORF translation elongation, the more efficient translation of the coding sequence. This would be at a cost, with the translation of the uORF sharply lowering translation of the coding sequence during non-stressed conditions (enhance approaching preferential translation).

The manuscript modeling also suggests a role for the strength of the translation initiation at the uORF. Too little initiation can thwart translation resistance. Furthermore, even modest reinitiation following translation of the uORF can block preferential translation of the coding sequence.

Overall, this manuscript addresses a timely and important question in the field: what are the mechanisms by which a single uORF can trigger preferential translation in the ISR. The presented models begin to lay out themes that can be applied genome-wide. Are the central ideas new to the literature? Some certainly are, such as the integration of ribosome scanning and elongation ribosomes in a computation model for the mechanism directing preferential or resistant translation in the ISR. However, the core ideas that elongation and different initiation events contribute to the translational control are plentiful in the literature. There are also some underlying assumptions to the computation model that are not as well supported in the literature This manuscript is computational, and key themes are not tested or even addressed significantly for correlations in the emerging seq- databases. These later concerns tempered enthusiasm for the manuscript.

*Reviewer #3:*

In this manuscript, the authors follow up on earlier work they published in *eLife* (Andreev et al., 2015). In the earlier paper they showed that some mRNA have increased translation under stress, and this required uORFs in their 5' UTRs. Here they explore possible explanations for this phenomenon (uORF-mediated stress resistance) using a computational model related to the TASEP modeling. TASEP (totally asymmetric simple exclusion process) modelling has been used previously to study how changing translation parameters through an annotated coding ORF (acORF as the authors say) affects protein production rates. This manuscript takes this one step further and uses a variation on TASEP to model scanning of 5' transcript leaders and translation of uORFs therein. This is an innovative approach to a difficult problem (understanding how various aspects of transcript leaders impact translation initiation), and the authors should be lauded for using this approach. After making certain assumptions (will return to this), their ICIER model suggests that part of uORF inhibition results from uORFs knocking off scanning PICs downstream. The authors model stress as reduced initiation of scanning, and find that transcript leaders with long uORFs have less repression under stress conditions than they do under non-stress. While interesting and thought provoking, the study is lacking in that some of the assumptions may not be valid, little detail is provided regarding the modeling, the results of the model appear reversed, compared to their previous work, and the figures are difficult to interpret. Computational modeling is always strongest when it can be used to make predictions that are affirmed by testing (e.g. with reporter constructs). The work would be much strengthened if the authors could present experimental data supporting the results of their model. Individual issues follow:

1) The authors results depend entirely on the assumption that elongating ribosomes remove scanning PICs when they collide, and scanning PICs don't affect each other or elongating ribosomes. This is why uORFs are more inhibitory at high rates of scanning initiation (*r_in_*). PICs that leak through the uORF start are stochastically stripped off by the PICs that don't leak through (and initiate). They say that Archer et al.'s results on translating complex profiling supports this assumption, but I don't really see that's the case from the Archer paper. It would be very helpful if the authors could thoroughly support this assumption with experimental evidence, as it underlies the rest of their results. Alternatively, please explain how the Archer et al. results fully support this crucial assumption.

2) They also assume that ribosomes translating uORFs do not affect each other in collisions and that upstream scanning PICs cannot remove uORF translating 80S ribosomes. I don't see how this is supported by Archer et al., and doesn't make intuitive sense to me. Please explain how Archer supports these assumptions.

3) The authors assume that PICs occupy the same footprint as 80S ribosomes, though Archer et al. finds this isn't the case. It may not matter, but should be evaluated.

4) The authors assume that there's no off-rate for scanning PICs or translating 80S ribosomes. The PICs are the only complex that falls off, and they only fall off when hit from behind by an 80S. This doesn't make sense, and the impact of PIC and 80S falloff should be evaluated.

5) The translation initiation process appears to be instantaneous in the ICIER model, though experimentally this seems to be a slow and rate-limiting step in translation (ribosome profiling reads tend to pileup on start codons). The model should include a pause at uORF initiation.

6) Overall, a Materials and methods section would have been very helpful as the text and figures don't have enough detail to fully understand the authors’ approach. I would also suggest posting their code to github or making it otherwise available to help reviewers and readers.

7) Their previous *eLife* paper used a spike-in control for ribosome profiling. This was a great innovation and allowed them to conclude that some mRNA are resistant to stress because they have translated uORFs. I expect that if they compared a transcript in their model with and without a uORF, the uORF-less one would still be more translated under stress than the uORF containing one. Yes, translation of a uORF containing gene increases (relatively speaking) under stress compared to its rate under non-stress, but this is still lower than if you just dropped the uORF altogether. Thus, I expect their model would show transcripts lacking uORFs are better translated under stress than those having uORFs.

8) Their model treats stress as a decrease in *r_in_* (PICs starting to scan). While this is plausible, it ignores the well-known effects of eIF2α phosphorylation under stress, which should lead to uORF skipping (decrease in *t_σ>ε_*) and acORF skipping. This is ignored in their model. How do the results change with changing *t_σ>ε_*?

9) Related to the above, it seems that a global decrease in acORF translation would provide a large increase in free 40S subunits, leading to an increase in *r_in_* PIC scanning and a decrease in *t_σ>ε_*for uORFs and acORFs.

10) Overall, some experimental data supporting the results of the model would help bolster the authors' interpretations. A global comparison to their published ribosome profiling data (and other public data) would be helpful, but really they need some reporter assays in which they vary the length of uORFs, Kozak context, elongation rate, etc. and see how they affect expression with and without stress. Combined with more thorough modeling, this might allow them to differentiate between changes in *r_in_* and *t_σ>ε_*under stress, and possibly support the assumption that scanning PICs can be removed by upstream translating 80S ribosomes (and not vice-versa). This would greatly strengthen the manuscript. Without experimental testing of model results, this is an interesting thought experiment suitable for theoretical bioinformatics journals, but whose results may not apply to living cells due to the assumptions used.

11) The y-axis scales for Figures 2-5 are very hard to interpret. Why is *r_out_* so much larger than *r_in_*? Shouldn't they be on the same scale, especially when they are in the same units (ribosomes/tact)? I believe I understand the percentages, which must be relative to the rate at *r_in_* = 0.1 (relative maximum increase), but the absolute ribosomes / tact seems off by many orders of magnitude (e.g. in Figure 4A).

12) Figure 4C and D are identical (looks like an error in figure prep).

In summary, while I really appreciate the interesting approach the authors take in modeling initiation rates with uORFs and feel that their results are interesting (when comparing effects of elongation rates, initiation rates, separately, e.g.), I believe they have made too many assumptions and ignored too many important aspects of the process to draw firm biological conclusions. With more thorough modeling and experimental testing, this could turn into a strong manuscript for e*Life*. At present however, it seems more suitable for a theoretical or bioinformatics journal (similar to another recent modeling paper the published in BMC Bioinformatics (Michel et al., 2014) that was in some ways superior in that the model results were consistent with public ribosome profiling data.

---

## [Author Response]

Reviewer #2:[…] The premise of the manuscript is that scanning 40S ribosomes dissociate from the 5'-leader of mRNAs when they proceed into (collide) with elongating ribosomes that are translating an uORF. In a sense, scanning ribosomal units can queue up behind an elongating ribosome. The interface between scanning and elongating ribosomes reflects the amount of the preinitiation ribosomal complex loading onto the 5'-end of the mRNAs and the efficiency of elongating ribosomes. In the case of DDIT3 there is a reported ribosomal stall, which could exacerbate the queue of scanning ribosomes. Other mechanisms, some linked to stress, may also lower translation elongation of the uORFs. The manuscript modeling suggests that the uORF length can change the likelihood of ribosomal collisions between scanning ribosomes and those elongating. Furthermore, eIF2 phosphorylation may alter the loading of ribosomes onto mRNAs and hence the flux of scanning ribosomes that can queue up. The manuscript modeling suggests that resistant mRNA translation of the coding sequence is enhanced when elongating ribosomes at uORFs are slight slower that the incoming scanning ones. The slower the uORF translation elongation, the more efficient translation of the coding sequence. This would be at a cost, with the translation of the uORF sharply lowering translation of the coding sequence during non-stressed conditions (enhance approaching preferential translation).

The reviewer correctly identified the key assumption of our modelling: that the scanning ribosome dissociates from the mRNA when it collides with elongating ribosomes although formulated it differently in his or her comments. The manuscript text: “We hypothesized that scanning ribosomes would dissociate from mRNA when upstream elongating ribosomes collide with them.” This is also clearly depicted in Figure 1A: The scanning ribosome (blue) is downstream of the elongating one (red) at the moment of dissociation. The reviewer’s interpretation or formulation is the other way around, i.e. that the scanning ribosomes dissociate when they collide with elongating ribosomes located downstream (“proceed into”). This is not the case in our model. In our model scanning ribosomes indeed queue as the referee suggested and the evidence for formation of such queues was provided by Archer et al., 2016 (see their Figure 4B) and we referenced that: “This hypothesis is based on the observation that scanning ribosomes queue behind elongating ribosomes and each other but the opposite has not been observed [Archer et al., 2016].”

The manuscript modeling also suggests a role for the strength of the translation initiation at the uORF. Too little initiation can thwart translation resistance. Furthermore, even modest reinitiation following translation of the uORF can block preferential translation of the coding sequence.Overall, this manuscript addresses a timely and important question in the field: what are the mechanisms by which a single uORF can trigger preferential translation in the ISR. The presented models begin to lay out themes that can be applied genome-wide. Are the central ideas new to the literature? Some certainly are, such as the integration of ribosome scanning and elongation ribosomes in a computation model for the mechanism directing preferential or resistant translation in the ISR. However, the core ideas that elongation and different initiation events contribute to the translational control are plentiful in the literature. There are also some underlying assumptions to the computation model that are not as well supported in the literature This manuscript is computational, and key themes are not tested or even addressed significantly for correlations in the emerging seq- databases. These later concerns tempered enthusiasm for the manuscript.

We thank the reviewer for highlighting the methodological novelty of our work. We agree that the idea that elongation and initiation events contribute to the stress response is not novel and we did not attempt to make such claim. To make it more apparent we extended our Introduction in these regards and changed the wording, e.g. instead “the model reveals derepression of downstream translation as general mechanism” we say “the model supports derepression of downstream translation as general mechanism”.

Further the reviewer mentioned that the core assumptions of the model are not supported by the literature. We are not sure to what extent this criticism is caused by most likely inadvertent misinterpretation of our assumptions (see our response earlier). However, in any case it may be fruitful to systematically consider all possible outcomes of ribosome collisions. From a purely theoretical point of view (irrespective of experimental evidence) we could consider six scenarios or a combination of them:

1) No dissociation takes place.

2) A scanning ribosome dissociates when it encounters a scanning ribosome downstream.

3) A scanning ribosome dissociates when it encounters an elongating ribosome downstream.

4) A scanning ribosome dissociates when an upstream scanning ribosome collides with it.

5) A scanning ribosome dissociates when an upstream elongating ribosome collides with it.

6) Dissociation of a scanning ribosome is an entirely stochastic process that is independent of their collisions with other ribosome complexes.

Archer et al. demonstrated formation of scanning ribosome queuing, therefore we could reject the possibilities 2 and 4 as no scanning ribosome queuing would be possible under these two scenarios. Further, it has been demonstrated just recently that an elongating ribosome pause can induce initiation at a remote location upstream which would require formation of a queue, see Ivanov et al., 2018.

This finding suggests that the scenario 3 does not occur either.

This leaves three options, 1, 5 and 6.

Option 1. If no dissociation takes place, under abstraction of our model the number of scanning ribosomes passing beyond the end of the uORF would equal the number of ribosomes not initiating at the uORF start codon. In other words, uORF would have no any effect on stress responses. Therefore, if scanning ribosomes do not dissociate we will have to assume that there are additional factors (currently unknown) that are responsible for the stress resistance. This is possible, however, in our reasoning we follow Ockham’s razor principle: when considering multiple plausible explanations, one should give preference to the simplest explanation that requires fewer unknowns.

Option 5. We consider this option as the one most likely reflecting the reality and performed most of our modelling under this condition.

Option 6. There is evidence to suggest that ribosomes do not dissociate spontaneously from mRNA. If they were, translation initiation would depend on the length of 5’ leaders which does not seem to be the case, e.g. Dmitriev et al., 2007. Nonetheless, we decided to consider this possibility and carried out simulations under this condition to see how the system behaves. As can be seen in Figure 6C (P_σ_ is the probability of spontaneous scanning ribosome dissociation) no stress resistance was observed as in the case of no dissociation (red line), the rate of scanning ribosome at the end of uORF is decreasing with increased probability of the scanning ribosome dissociation (as expected).

We also combined option 5 and option 6 (Figure 6A), where scanning ribosomes dissociate when they collide with elongating ribosomes upstream (red line), but also could spontaneously dissociate with certain probabilities (P_σ_). It appears that such spontaneous dissociations reduce stress resistance (which is also expected).

We now explain our reasoning for the support of option 5 in the revised manuscript, we also provide results of the simulations under assumptions of options 1, 6 and a combination of 5 and 6 in new Figure 6.

Further the reviewer suggested to test the predictions of our model using emerging public datasets. We thank the reviewer for this suggestion. To address this, we chose the following ribosome profiling datasets where Integrated Stress Response pathway was induced Andreev et al., 2015, Sidaruski et al.,

2015 and Oh et al., 2016. In summary we explored how length of uORFs, translation initiation efficiency (approximated as ribosome footprint density) and the presence of pauses affected the stress resistance. The length of uORF is supported the most strongly in all three studies (Figure 7A). The influence of initiation is supported by Andreev et al. while Sidaruski and Oh do not exhibit such correlations (Figure 7B). There is a weak positive correlation between the presence of pauses and stress resistance in all studies, however, it is not statistically significant (Figure 7C). We discuss these findings in the manuscript.

Reviewer #3:[…] 1) The authors results depend entirely on the assumption that elongating ribosomes remove scanning PICs when they collide, and scanning PICs don't affect each other or elongating ribosomes. This is why uORFs are more inhibitory at high rates of scanning initiation (r_in_). PICs that leak through the uORF start are stochastically stripped off by the PICs that don't leak through (and initiate). They say that Archer et al.'s results on translating complex profiling supports this assumption, but I don't really see that's the case from the Archer paper. It would be very helpful if the authors could thoroughly support this assumption with experimental evidence, as it underlies the rest of their results. Alternatively, please explain how the Archer et al. results fully support this crucial assumption.

The”collision assumption” comment is parallel to that of reviewer 2, please see our response to the corresponding comment above. Also note that we understand that the real situation is likely to be much more complicated, i.e. if ribosomes dissociate they would not do it immediately at the moment of the collision and even when there is evidence for queuing as among scanning ribosomes (see their Figure 4B Archer et al., 2016), it does not mean that such ribosomes do not dissociate. However, the mathematical modelling necessitates certain level of abstraction. Such abstraction indeed may affect our ability to make quantitative predictions, but at this point our goal is to gain insight and model qualitative behavior. Our main finding in this regard is that a very simple mathematical model (very abstract and striped off many details) is able to explain how a single uORF can lead to stress resistance.

2) They also assume that ribosomes translating uORFs do not affect each other in collisions and that upstream scanning PICs cannot remove uORF translating 80S ribosomes. I don't see how this is supported by Archer et al., and doesn't make intuitive sense to me. Please explain how Archer supports these assumptions.

In this comment the reviewer suggest that scanning ribosomes can remove elongating ribosomes. While we cannot rule out such a possibility completely it seems highly unlikely for the following reasons:

1) It is difficult to dissociate elongating ribosome form mRNA, even when peptydl hydrolysis takes place at the stop codon during termination, i.e. splitting ribosomes requires ABCE1. Isolation of mRNA complexes with multiple elongating ribosomes bound to it is a routine procedure. Stabilization of scanning ribosomes with mRNA requires chemical cross-linking.

2) If such dissociations were taking place, we would expect a large number of polypeptides truncated at the C-terminus. We are not aware of evidence supporting this.

3) Pretreatment of lysates with cyclheximide blocks elongating ribosomes on mRNAs, but not scanning ribosomes. If scanning ribosomes were removing elongating ribosomes, we would observe depletion of ribosome density at the 5’ ends. However, the exact opposite is observed in what is now considered to be a technical artefact (nicely explained by Jackson and Standart, 2015).

4) If scanning ribosomes were removing elongating ribosomes then the queue of scanning ribosomes observed by Archer et al. would not form. Also the stimulation of initiation by stalled elongating ribosomes reported recently (Ivanov et al., 2018) would not be possible.

3) The authors assume that PICs occupy the same footprint as 80S ribosomes, though Archer et al. finds this isn't the case. It may not matter, but should be evaluated.

We evaluated the effect of scanning footprints length as suggested by the reviewer. To our surprise, it appears that at least in our model, the length of scanning ribosomes has a dramatic effect on stress resistance and inhibition (see new Figure 6D where D is the length of scanning ribosome in codons). Smaller scanning ribosomes increase the inhibitory effect of uORF and subsequently provide higher resistance. Most likely this is due to the reduced frequency of ribosome collisions. While it is hard to assess how it relates to the reality, it helps to understand the model better, it also highlights the importance of this parameter for quantitatively accurate modeling, hence we thank the reviewer for this suggestion.

4) The authors assume that there's no off-rate for scanning PICs or translating 80S ribosomes. The PICs are the only complex that falls off, and they only fall off when hit from behind by an 80S. This doesn't make sense, and the impact of PIC and 80S falloff should be evaluated.

For the reasons why we think that elongating ribosomes remove scanning once, see our response to the reviewer 2 “collision comment” earlier. Regarding elongating ribosomes drop-off, there is sufficient evidence suggesting that the rate of elongating ribosomes is negligible, and that it may occur with noticeable efficiency only at particular circumstances (either sequence specific, on particular mRNA location, or under very specific cellular conditions). If we consider the ribosome as universal protein synthesis machinery operating in every cell (alternative hypothesis to consider is specialized ribosomes), it is impossible to explain how full length 38138 aa long Titin can be ever translated with ability of 80S to falloff with any significant rate. The initial ribosome profiling experiments failed to detect any drop-off in mammals (Guo et al., 2010) and very little drop-off in bacteria (Oh et al., 2011). This has been revisited later (Sin et al., 2016), however, the level of detected drop-off is still very small ~10^-4^ events per codon in *E. coli*. Note, that it is likely to be even lower in eukaryotes since some ORFs are much longer in eukaryotes. Nonetheless we explored how different levels of elongating ribosome drop-off affect our model, see new subsection “Miscellaneous parameters of the model: ribosome gabarits and fall off rates”.

Scanning ribosomes, on the other hand, definitely can fall off without collision with elongating ribosome (otherwise all uORFless mRNAs with a start codon in the good context would be translated with same efficiency), but how and why it happens is not clear, so we can’t properly model this. For example, there is no clear relationship between 5’ leader length and translation efficiency, arguing against constant low rate fall off (Dmitriev at al., 2007). It seems that there are some local mRNA 5’ leader features (primary or secondary RNA structure, RNA binding protein) that can induce scanning ribosome fall off. Therefore, we agree with the reviewer and we carried out simulations under the conditions where scanning ribosome falls at different rates, either in combination with dissociations caused by collisions or not, see new subsection “Miscellaneous parameters of the model: ribosome gabarits and fall off rates”.

5) The translation initiation process appears to be instantaneous in the ICIER model, though experimentally this seems to be a slow and rate-limiting step in translation (ribosome profiling reads tend to pileup on start codons). The model should include a pause at uORF initiation.

We agree. We introduce such a parameter in the model and explored how such pausing affects stress resistance. We expanded the subsection “Strength and pausing of translation initiation at uORF and probability of re-initiation downstream” accordingly.

6) Overall, a Materials and methods section would have been very helpful as the text and figures don't have enough detail to fully understand the authors’ approach. I would also suggest posting their code to GitHub or making it otherwise available to help reviewers and readers.

We expanded the methods and deposited the code to GitHub: https://github.com/maximarnold/uORF_TASEP_ICIER

7) Their previous eLife paper used a spike-in control for ribosome profiling. This was a great innovation and allowed them to conclude that some mRNA are resistant to stress because they have translated uORFs. I expect that if they compared a transcript in their model with and without a uORF, the uORF-less one would still be more translated under stress than the uORF containing one. Yes, translation of a uORF containing gene increases (relatively speaking) under stress compared to its rate under non-stress, but this is still lower than if you just dropped the uORF altogether. Thus, I expect their model would show transcripts lacking uORFs are better translated under stress than those having uORFs.

Yes, the reviewer is correct. We included uORFless simulation that clearly shows that.

8) Their model treats stress as a decrease in r_in_ (PICs starting to scan). While this is plausible, it ignores the well-known effects of eIF2α phosphorylation under stress, which should lead to uORF skipping (decrease in t_σ>ε_) and acORF skipping. This is ignored in their model. How do the results change with changing t_σ>ε_?

We are not sure to what exactly the reviewer refers to as uORF skipping. In classical Alan Hinnebiusch GCN4 example it is the long downstream uORF that is skipped. This happens because the scanning ribosome that emerges after termination at the upstream uORF (reinitiate) temporarily lacks ternary complex (as under condition of eIF2 phosphorylation the concentration of ternary complex is greatly reduced). There is no evidence that PIC can start to scan from 5’end of the mRNA without eIF2.

9) Related to the above, it seems that a global decrease in acORF translation would provide a large increase in free 40S subunits, leading to an increase in r_in_ PIC scanning and a decrease in t_σ>ε_ for uORFs and acORFs.

Under the stress the limiting factor is the ternary complex, not 40S. In the absence of the ternary complex, PIC could not be formed. We cannot exclude a possibility that other initiation factors (for example, eIF3 or eIF4F) availability increases under stress and this leads to increase in *r_in_* on certain 5’ leaders, where these factors are rate limiting under normal conditions. This is however hard to take into account during TASEP based modelling.

10) Overall, some experimental data supporting the results of the model would help bolster the authors' interpretations. A global comparison to their published ribosome profiling data (and other public data) would be helpful, but really they need some reporter assays in which they vary the length of uORFs, Kozak context, elongation rate, etc. and see how they affect expression with and without stress. Combined with more thorough modeling, this might allow them to differentiate between changes in r_in_ and t_σ>ε_ under stress, and possibly support the assumption that scanning PICs can be removed by upstream translating 80S ribosomes (and not vice-versa). This would greatly strengthen the manuscript. Without experimental testing of model results, this is an interesting thought experiment suitable for theoretical bioinformatics journals, but whose results may not apply to living cells due to the assumptions used.

We agree with the reviewer that further experimental investigation of the matter is needed. Experimental testing of all the possible parameters is a major endeavor that would require substantial time and funding, as we can’t rely on single cellular mRNA 5 leader. In this case, one would argue that there are unique features such as specific codons in the uORF that influence stress response. Because of this we would have to vary many parameters in context of dozens of 5’leaders bearing a single uORF. If one were to do it properly, this would require generation of hundreds of reporter constructs and thousands of experiments. We therefore, believe, it would warrant a separate work in the future. We hope that the data obtained from such work would allow us to tune the parameters of the model and achieve quantitative rather than qualitative predictions.

11) The y-axis scales for Figures 2-5 are very hard to interpret. Why is r_out_ so much larger than r_in_? Shouldn't they be on the same scale, especially when they are in the same units (ribosomes/tact)? I believe I understand the percentages, which must be relative to the rate at r_in_ = 0.1 (relative maximum increase), but the absolute ribosomes / tact seems off by many orders of magnitude (e.g. in Figure 4A).12) Figure 4C and D are identical (looks like an error in figure prep).

We thank the reviewer for spotting this problem. These values were obtained for a large number of simulations and we forgot to divide these rates by the number of simulations. Corrected now.